# Amphetamine Exposure during Embryogenesis Alters Expression and Function of Tyrosine Hydroxylase and the Vesicular Monoamine Transporter in Adult *C. elegans*

**DOI:** 10.3390/ijms25084219

**Published:** 2024-04-11

**Authors:** Tao Ke, Katie E. Poquette, Sophia L. Amro Gazze, Lucia Carvelli

**Affiliations:** 1Harriet L. Wilkes Honors College, Florida Atlantic University, Jupiter, FL 33458, USAkpoquette2020@fau.edu (K.E.P.); samrogazze2020@fau.edu (S.L.A.G.); 2Stiles-Nicholson Brain Institute, Florida Atlantic University, Jupiter, FL 33458, USA

**Keywords:** amphetamine, tyrosine hydroxylase, VMAT, histone methylation, *C. elegans*

## Abstract

Amphetamines (Amph) are psychostimulants broadly used as physical and cognitive enhancers. However, the long-term effects of prenatal exposure to Amph have been poorly investigated. Here, we show that continuous exposure to Amph during early development induces long-lasting changes in histone methylation at the *C. elegans* tyrosine hydroxylase (TH) homolog *cat-2* and the vesicular monoamine transporter (VMAT) homologue *cat-1* genes. These Amph-induced histone modifications are correlated with enhanced expression and function of CAT-2/TH and higher levels of dopamine, but decreased expression of CAT-1/VMAT in adult animals. Moreover, while adult animals pre-exposed to Amph do not show obvious behavioral defects, when challenged with Amph they exhibit Amph hypersensitivity, which is associated with a rapid increase in *cat-2*/TH mRNA. Because *C. elegans* has helped reveal neuronal and epigenetic mechanisms that are shared among animals as diverse as roundworms and humans, and because of the evolutionary conservation of the dopaminergic response to psychostimulants, data collected in this study could help us to identify the mechanisms through which Amph induces long-lasting physiological and behavioral changes in mammals.

## 1. Introduction

Amphetamine (Amph) is a psychostimulant that has been used to treat a variety of brain dysfunctions, but it is accompanied by a high abuse liability. As a matter of fact, Amph and Amph-derived compounds, such as methamphetamine (Meth), are among the most abused psychostimulants worldwide. The neurological effects caused by acute or chronic administrations of Amph have been broadly investigated [1], and several studies have shown that proteins involved in the synthesis, storage, release, and reuptake of dopamine (DA) are either direct targets of or are indirectly affected by these drugs. For example, following acute or chronic treatments, Amph increases extracellular DA by directly interacting with the DA transporter (DAT) [2]. The interaction of Amph with DAT prevents DAT-mediated DA uptake and, at the same time, allows rapid access of Amph into the neurons. Once inside the neurons, Amph interacts with the vesicular monoamine transporter (VMAT), which transports monoamines, including DA, into the synaptic vesicles. The binding of Amph with VMAT prevents accumulation of DA in the vesicles and causes redistribution of DA from the vesicles to the cytoplasm by inducing deacidification of the vesicles via a VMAT-mediated H^+^ antiport [3]. Interestingly, chronic treatments with Amph also cause a 70% decrease in VMAT2 expression in postnatal rats [4], suggesting that exposure to Amph during early development modifies expression of specific dopaminergic genes. 

Tyrosine hydroxylase (TH), the rate-limiting enzyme of DA synthesis, is another important dopaminergic protein whose function is affected by Amph. Previous data suggested that there is an increase in DA synthesis during acute Amph treatments [1]. In fact, blockage of TH reduced the stimulant effects caused by Amph in mice [5]. This result was supported by other data showing that acute Amph treatments increased the activity of TH in rats [6]. Interestingly, though, chronic treatments had a different outcome on TH activity. In fact, Park et al. (2012) reported a decrease in TH expression after chronic treatments with Meth [7], and a reduction in TH expression was also observed in adult mice after receiving binge-like exposure to Meth at a younger age [8]. On the other hand, two studies reported that adult rats that underwent prenatal Meth exposure exhibited higher levels of basal DA and higher responses to challenging doses of methamphetamine [9]. These results indicate that more studies are needed to better understand the long-term consequences of chronic Amph exposure, particularly during early development. Nonetheless, the existing data suggest that continuous exposure to Amph during early development, when neuronal wiring is established, could interfere with basal neuronal sprouting, and the Amph-induced changes could then cause functional and behavioral alterations in adult animals [10]. In other words, continuous exposure to Amph during early development can cause functional changes that persist in the absence of the drug. However, the mechanism(s) underlying these long-lasting effects have not yet been elucidated. 

Recent human studies have shown that in utero exposure to low/therapeutic doses of Amph used to treat attention deficit disorders does not affect the overall health of the offspring [11]. On the other hand, animal studies demonstrated that prenatal exposure to high/addictive concentrations of Meth caused elevated levels of DA and higher behavioral responses to Meth in adult animals [9]. Yet, no study has investigated the mechanisms underlying the long-lasting effects seen after in utero exposure to high concentrations of Meth or Amph. Using *C. elegans*, we recently demonstrated that adult animals exposed to high concentrations of Amph during embryogenesis exhibit reduced levels of DAT and are hypersensitive to Amph [12]. We also demonstrated that the long-lasting effects caused by Amph are mediated by changes of histone marks at the *dat-1* gene. This opens the question of whether embryonal Amph also changes the expression and function of proteins involved in DA synthesis and storage. Since *C. elegans* express highly conserved human TH and VMAT homolog genes, *cat-2* and *cat-1* respectively, we investigated whether Amph exposure during early development affects the expression and/or function of TH/CAT-2 and VMAT2/CAT-1 in adult *C. elegans* via epigenetic mechanisms.

## 2. Results

### 2.1. Exposure to Amphetamine during Embryogenesis Alters Histone Marks in the C. elegans Tyrosine Hydroxylase Cat-2 Gene

Tyrosine hydroxylase (TH) is the rate-limiting enzyme in catecholamine biosynthesis. In mammals’ brains, it is expressed in neurons producing DA and/or norepinephrine. Previous reports demonstrated that Amph-like drugs affect TH activity. For instance, Larsen et al. showed that Meth increases cytosolic DA by upregulation of TH [13]. Moreover, data collected from animals, including *C. elegans*, showed that a functional TH is needed to generate Amph-induced behaviors [14,15]. Specifically, Heusner et al. showed that mice lacking the TH gene have a blunted response to Amph, and reinstatement of DA production by adeno-associated viruses expressing TH in the nucleus accumbens completely restored their ability to respond to Amph [14]. However, little information is available about the functional consequences of prenatal exposure to Amph on TH activity in adult animals. One study by Bubenikova-Valesova et al. demonstrated that adult rats exposed prenatally to Meth have elevated levels of basal DA in the nucleus accumbens and elevated behavioral responses during Meth challenge [9]. This study, however, did not investigate whether the elevated basal levels of DA are the result of an upregulation of TH expression/activity or reduced degradation of DA. Moreover, the mechanisms that maintain, in adult animals, the effects of Amph initiated during embryogenesis are currently unknown.

Various epigenetic mechanisms, including histone changes and DNA methylation, have been involved in the long-lasting effects caused by drugs of abuse [16]. Since histone methylation is a crucial step in regulating gene expression during embryogenesis [17], we investigated whether Amph exposure during embryogenesis changes histone methylation in the *cat-2* gene of adult *C. elegans*. To do so, chromatin immune precipitation (ChIP) experiments for two specific histone marks, the histone 3 (H3) lysine 4 (K4) three-methylated (H3K4me3) and H3K9 dimethylated (H3K9me2), were performed. These histone marks were previously shown to be altered by psychostimulants in specific genes of the reward system both in mammals and *C. elegans* [12,18,19]. There are three alternative transcription start sites (TSSs) for the *cat-2* gene. We designed primer pairs 1, 2, and 3 at the promoters for each of these TSSs. Primers for site 1 are upstream of the TSS of the longest transcript, site 2 is at the TSS of transcript 2, and site 3 is immediately upstream of the TSS of transcript 3 (Figure 1a). Our ChIP results show a significant increase in H3K4me3 (* *p* = 0.0001, 2-way ANOVA) between control- and Amph-treated samples at site 3 (from 11.7 ± 0.9 to 20.4 ± 1.3) but not at site 1 (from 10.7 ± 0.8 to 10.38 ± 0.2) or 2 (from 12.1 ± 0.8 to 11.8 ± 0.4) of the *cat-2* gene (Figure 1b), whereas no change between control- and Amph-treated samples was measured at sites 1 or 2 of the housekeeping gene *gpdh* (from 11.4 ± 0.7 to 13 ± 0.5 and from 10.6 ± 0.3 to 11.3 ± 0.3, respectively) (Figure 1b), suggesting, therefore, that the histone changes caused by Amph are directed at specific genes. When we investigated the H3K9me2 mark, we found a dramatic reduction at sites 1 (from 14.8 ± 09 to 2.6 ± 0.7) and 3 (from 16.8 ± 0.9 to 3.1 ± 0.2), but not at site 2 (from 2.8 ± 0.6 to 3.1 ± 0.3) in the *cat-2* gene of adult animals treated with Amph during embryogenesis with respect to animals pretreated with control solution (Figure 1c; * *p* < 0.0001; 2-way ANOVA). Again, no change was measured at sites 1 or 2 of the *gpdh* gene which, here, is used as a control gene (from 3.8 ± 0.9 to 3.4 ± 0.9 and from 3.1 ± 0.3 to 2.9 ± 0.3, respectively, in Figure 1c). Taken together, these results suggest that exposure to Amph during early development rearranges the epigenetic landscape in the *cat-2* gene. Moreover, because H3K4me3 is associated with gene activation, whereas H3K9me2 is associated with gene repression [20,21], these data indicate that continuous exposure to Amph during embryogenesis would increase the expression of *cat-2* in adult animals.

### 2.2. Exposure to Amphetamine during Embryogenesis Upregulates Protein Expression of CAT-2/TH in Adult Animals

Our ChIP data (Figure 1b,c) show that Amph given during early development increases the activator mark H3K4me3 but decreases the inhibitor H3K9me2 in the *cat-2* gene, suggesting, therefore, that adult animals that received Amph during embryogenesis might have increased levels of CAT-2 proteins. We tested this hypothesis by performing Western blot experiments using whole-worm protein lysates. We found a significant increase in CAT-2 in samples extracted from young adults that went through embryonic Amph exposure with respect to animals exposed to control solution (from 100 ± 16 to 420 ± 64. * *p* = 0.02, paired t-test; Figure 2a). The same samples probed with the antibody against actin did not show a change in band intensity and/or size (inset in Figure 2a). Increased levels of CAT-2 (TH) would suggest higher levels of DA. To test if this was the case, we quantified DA in whole-worm extracts by performing ELISA assays using a specific antibody against DA. We found a statistically significant increase in DA (from 1.3 ± 0.1 to 2.4 ± 0.2) in samples from adult animals exposed to Amph during embryogenesis with respect to animals exposed to control solution (Figure 2b; * *p* = 0.0003; paired *t*-test). Taken together, these results demonstrate that embryos exposed to Amph develop into adult animals that express higher protein levels of TH (CAT-2) and produce more DA than control animals. To note, these data are in alignment with our epigenetic data supporting an increase in expression of *cat-2*/TH (Figure 1b,c) and with previous studies from murine models showing that prenatal or chronic treatments with Amph caused a significant increase in DA in the striatum [9,22]. The resemblance between the murine and our *C. elegans* data suggest that the physiological response to Amph is highly conserved across species.

Recently, using a DA-dependent behavioral assay named SWIP (Swimming-Induced Paralysis), we showed that exposure to Amph during embryogenesis causes hypersensitivity to Amph in adult *C. elegans* [12]. SWIP is an easily quantifiable behavior caused by an excess of extracellular DA. While in water, *C. elegans* swim vigorously; but in solutions containing a DA releaser, e.g., Amph, or an inhibitor of the dopamine transporter, e.g., mazindol, which increases extracellular DA, *C. elegans* stop swimming and exhibit SWIP [15,23]. Thus, SWIP can be used as a readout of extracellular DA overflow [15,24]. Since our data show increased levels of TH and DA in animals exposed to Amph during embryogenesis, we investigated whether these animals exhibit differences in basal and/or Amph-induced SWIP. We found that, similarly to animals pretreated with control solution, adult animals exposed to 0.5 mM Amph during embryogenesis do not exhibit SWIP when tested in vehicle (Figure 2c, black circles and green triangle, respectively), suggesting that embryonal Amph treatments do not affect basal DA-dependent behaviors. However, when animals were challenged with 0.5 mM Amph for 10 min, we observed significantly increased SWIP in animals exposed to Amph during embryogenesis with respect to animals pre-exposed to control solution (Figure 2c, red and blue circles, respectively). These results confirm our previous reports [12] and resemble recent data showing that mice overexpressing TH exhibited enhanced sensitivity to Amph and elevated striatal DA levels, but no basal hyperdopaminergic behaviors [25,26]. To further investigate the Amph hypersensitivity caused by embryonal exposure to Amph, we quantified *cat-2* mRNA in adult animals exposed to Amph during embryogenesis before and after the 10 min of Amph challenge. We found no difference in *cat-2* mRNA expression between adult controls and Amph-exposed animals during embryogenesis after animals swam for 10 min in vehicle (from 1 ± 0.1 to 0.8 ± 0.2; Figure 2d). Interestingly, though, after animals swam in 0.5 mM Amph for 10 min, we found a significant increase in *cat-2* mRNA (from 1 ± 0.05 to 4.1 ± 1.2) in animals exposed to Amph during embryogenesis with respect to those exposed to control (Figure 2d; * *p* = 0.01, 2-way ANOVA). This result suggests that after exposure to Amph during early development, adult animals respond to acute treatments with Amph by rapidly increasing expression of *cat-2* mRNA. 

### 2.3. Exposure to Amphetamine during Embryogenesis Alters Histone Marks and Expression of the VMAT2 Homolog Cat-1 Gene in Adult C. elegans

The vesicular monoamine transporter 2 (VMAT2) is responsible for packaging monoamines, such as DA, in synaptic vesicles. This guarantees the quantum release of neurotransmitters in the synaptic cleft after Ca^2+^-stimulated vesicle release. Previous studies showed that Amph is a direct substrate of VMAT2 [2], and studies from *Drosophila melanogaster* showed that Amph-induced behaviors require a functional VMAT2 [3]. Similarly, in *C. elegans*, Amph-induced SWIP is reduced by 50% in mutants lacking expression of the *C. elegans* VMAT2 homolog, *cat-1* [15]. Moreover, using murine models, several labs have shown that changes in VMAT gene expression alter DA signaling. For example, Lohoff et al. (2019) demonstrated that the frontal cortex of VMAT1 KO mice exhibits decreased DA levels, increased expression of the postsynaptic D2 receptors, and lower TH expression with respect to wild-type mice [27]. Importantly, VMAT1 KO mice also display an exaggerated locomotor response to acute Amph treatment. Here, we investigated if embryonic exposures to Amph during embryogenesis cause histone modifications at the *cat-1*/VMAT2 gene and, thus, changes in CAT-1/VMAT2 expression in adult *C. elegans*. Primers for the ChIP assays were designed to detect histone marks upstream of (1) and at the (2) transcription start site, as well as within the second intron (3) of the *cat-1* gene (Figure 3a). 

We found that adult animals that were treated with Amph during embryogenesis exhibited a significant reduction, from 6.6 ± 0.6 to 4.1 ± 0.1, of H3K4me3 at site 1 (Figure 3b; * *p* = 0.004; 2-way ANOVA). No significant difference was found for the *cat-1* sites 2 (from 6.8 ± 0.8 to 5.2 ± 0.6) or 3 (from 5.7 ± 0.7 to 4.1 ± 0.3). Using the same set of primers, we found a robust increase in the H3K9me2 mark (* *p* < 0.0001; 2-way ANOVA) in site 1 (from 0.3 ± 0.05 to 2.2 ± 0.3), site 2 (from 0.2 ± 0.02 to 2.3 ± 0.5), and site 3 (form 0.3 ± 0.08 to 1.51 ± 0.4) of the *cat-1* gene (Figure 3c). And, as seen with *cat-2*, the long-lasting epigenetic effects of Amph were confirmed to be specific for the dopaminergic genes (*cat-1* and *cat-2*) because no change in H3K4me3 or H3K9me2 was measured for the housekeeping gene *gpdh* (Figure 3b,c). Of note, and as expected, *gpdh* has high levels of H3K4me3 and low levels of H3K9me2 (Figure 1b,c and Figure 3b,c), which is consistent with the high levels of expression of a housekeeping gene. The lack of effect by Amph in the *gpdh* gene suggests that the long-lasting effects caused by embryonal exposure to Amph do not induce a widespread effect, but it occurs in specific dopaminergic genes (*cat-2* and *cat-1*) only. Taken together, these results show that Amph exposure during embryogenesis decreases the activator mark H3K4me3 and dramatically increases the silencing mark H3K9me2 at the *cat-1* gene, suggesting, therefore, that embryonic exposure to Amph hampers the expression of *cat-1* in adult animals. Indeed, when we quantified *cat-1* gene expression, we found that adult animals exposed to Amph during embryogenesis (Embryo-Amph) exhibited a 27 ± 0.08% reduction in *cat-1* mRNA with respect to animals pre-exposed to control solution (Figure 3d; *****
*p* = 0.03; *t*-test). Similarly, Western blot assays showed a significant reduction in GFP-fused CAT-1 proteins in samples exposed to Amph during embryogenesis with respect to control-exposed animals (Figure 3e; * *p* = 0.02; *t*-test). Taken together, these results demonstrate that exposure to Amph during embryogenesis decreases expression of *cat-1* mRNA and CAT-1 proteins in young adults.

### 2.4. The Behavioral Effects Caused by Embryonic Amphetamine Exposure Are in Part Blocked by Genetic Inhibition of MET-2, the C. elegans Homolog of the Human Histone 3 Lysine 9 Methyltransferase SETDB1

Recent studies have shown that H3K9me2-mediated silencing is an important epigenetic mechanism of gene expression throughout development [28]. Interestingly, our ChIP data show dramatic changes in H3K9me2 at both *cat-2* and *cat-1* genes in adult animals exposed to Amph during early development (Figure 1c and Figure 3c), suggesting that H3K9me2 plays a prominent role in the long-lasting effects caused by Amph. Methylation of lysine 9 (K9) in histone 3 (H3) is accomplished by specific enzymes broadly known as H3K9 methyltransferases. Five H3K9-specific methyltransferases have been identified in mammals [29], and two of them, G9A and SETDB1, have *C. elegans* homologs, SET-25 and MET-2, respectively [28]. Recently, we showed that pharmacologic and genetic ablation of SET-25 eliminated the long-lasting behavior effects caused by embryonic exposure to Amph [12]. Here, we tested if ablation of the *met-2* gene would generate similar results. We found that, contrarily to wild-type animals, *met-2* knockouts did not exhibit Amph hypersensitivity following Amph exposure during embryogenesis (Figure 4). In fact, no change was measured in Amph-induced SWIP in the group of animals that received Amph during embryogenesis with respect to the group pretreated with control solution (compare red and blue circles in Figure 4 with red and blue circles in Figure 2c). And, as seen in the wild-type animals (Figure 2c), no change in SWIP was recorded between the two groups challenged with vehicle solutions (Figure 4, green triangles and black circles). These results might suggest that loss of function of the H3K9 methyltransferase MET-2 prevents the long-lasting behavioral effects caused by exposure to Amph during embryogenesis. However, when we compared *met-2* knockouts with wild-type animals, we observed that, even in the absence of Amph pre-exposure, the *met-2* mutants exhibited significantly higher SWIP after Amph challenge with respect to wild-type animals (compare blue circles in Figure 2c and Figure 4; * *p* < 0.0002, 2-way ANOVA, Tukey’s multiple comparisons test). Since MET-2 is predominantly expressed in the cytoplasm rather than in the nucleus [30], it is possible that MET-2 methylates non-nucleosomal histones, as previously shown for its human homolog SETDB1 [31,32]. Thus, the increased Amph-induced SWIP seen in the *met-2* mutants could be the result of methylation defects in not-yet-identified cytoplasmic proteins.

## 3. Discussion

While the effects of therapeutical doses of amphetamine (Amph) taken during pregnancy have been investigated on birth outcomes in humans [11], a thorough investigation of the mechanisms underlying the long-term effects caused by embryonal exposure to addictive doses of Amph remains largely unexplored. Here, we investigated whether exposure to high doses of Amph throughout embryogenesis causes changes in the expression and function of two major dopaminergic proteins, TH and VMAT. The dopaminergic response to Amph and the mechanisms underlying histone methylation are highly conserved across diverse species [15,28]; thus, we used *C. elegans* to examine the long-term effects caused by embryonal exposure to Amph. One advantage of this model is that *C. elegans* embryos can develop outside the uterus and in the absence of maternal care. Therefore, our results were not influenced by possible Amph-induced epigenetic or behavioral modifications passed through maternal care [33,34], but they are a direct consequence of biological alterations at the embryo.

Our expression data show a reduction in CAT-1/VMAT2 in adult animals exposed to Amph during embryogenesis (Figure 3). In these animals, we can assume that reduced levels of CAT-1/VMAT2 cause an increase in cytosolic DA, which, in turn, induces the DA transporter (DAT) to work in reverse, moving DA outside the neuron [2]. This generates an increase in extracellular DA and, thus, higher levels of SWIP as seen in Figure 2c. Interestingly, though, the increase in extracellular DA, and thus the increased SWIP seen in the Amph-pretreated animals, appeared only after 3–10 min of acute Amph (Figure 2c, compare red with blue circles). Without the Amph challenge (vehicle groups), no difference in SWIP was recorded in animals pretreated with Amph with respect to those pretreated with control solution (Figure 2c, green triangles and black circles). This suggests that animals adapt to lower levels of CAT-1/VMAT under basal conditions. But, under Amph challenge, the DA overflow induced by Amph combined with the higher concentrations of cytosolic DA causes a surge of DA, which we detected in our behavioral assay as an increase in SWIP (Figure 2c, red circles).

Previously, we showed that, similarly to CAT-1/VMAT2, the *C. elegans* DAT-1 was also downregulated following Amph exposure during embryogenesis [12]. Thus, it is reasonable to assume that adults exposed to Amph during embryogenesis recycle DA less efficiently than control animals and, therefore, they should produce more DA. Indeed, we found that TH/CAT-2 was upregulated in adult animals exposed to Amph during embryogenesis (Figure 2a), and since DA was increased as well (Figure 2b), we speculate that the higher expression of *cat-2* is responsible for the increased DA levels. Interestingly, our gene expression data also show that adult animals exposed to Amph during embryogenesis have increased levels of *cat-2*/TH mRNA after 10 min of challenge with Amph (Figure 2d), suggesting that embryonic Amph exposure primes the *cat-2* gene for rapid expression when animals encounter Amph later in life.

Our behavioral data (Figure 2c) show that, following embryonic exposure to Amph, adult animals exhibit an increased response to Amph. This suggests that the altered expression of CAT-2/TH and CAT-1/VMAT caused by continuous exposure to Amph during embryogenesis generates animals that are hypersensitive to Amph. One may argue that 0.5 mM Amph could cause toxicity in *C. elegans*. However, after 15 h exposure with 0.5 mM Amph during embryogenesis, all *C. elegans* embryos develop in adults with no apparent disfunction and with intact dopaminergic neurons. In support of our data, Schreiber and McIntire (2010) showed that after a 16 h treatment with methamphetamine, only concentrations equal to or higher than 8 mM were toxic in *C. elegans* [35]. Moreover, like mammals, *C. elegans* express monoamine oxidases [36]; thus, we assume that in our experiments, Amph undergoes partial degradation. This suggests that the final concentration of Amph in *C. elegans*’ body is lower than 0.5 mM. 

Nucleosome rearrangement influences gene transcription, and, indeed, our data show that, following Amph exposure during embryogenesis, changes in methylation of H3K9 and, in part, H3K4 (Figure 1 and Figure 3b,c) are associated with changes in *cat-1* and *cat-2* expression (Figure 2a,d and Figure 3d,e). In alignment with our expression data, our ChIP data show a dramatic change of the repression mark H3K9me2 and a moderate change of the activation mark H3K4me3 in the *cat-2*/TH and *cat-1*/VMAT2 genes of adult animals following pre-exposure to Amph (Figure 1 and Figure 3b,c). For this reason, we focused our investigation on one of the two enzymes that in *C. elegans* add methyl groups to H3K9 and, thus, generate H3K9me2 [30]. We found that animals lacking expression of *met-2*, which is the *C. elegans* homolog of the mammalian SETDB1, did not exhibit the increased Amph response caused by embryonal exposure to Amph (compare Figure 2c and Figure 4). However, we also observed an increase in the basal response to Amph in the *met-2* mutants with respect to wild-type animals (compare blue circles in Figure 2c and Figure 4). At this point, we cannot explain why knocking down *met-2* would increase the acute response to Amph, but we can assume that, because MET-2 is primarily expressed in the cytoplasm rather than the nucleus [30], it is possible that lack of expression of MET-2 in the cytoplasm prevents methylation of non-nucleosomal proteins and this, in turn, alters the basal response to Amph. Future experiments will determine the nature of these cytoplasmatic proteins.

In summary, our data show that continuous exposure to Amph during embryogenesis increases CAT-2/TH and decreases CAT-1/VMAT expression in adult *C. elegans* via epigenetic mechanisms. Considering that our results are in agreement with data showing that mice overexpressing TH exhibit enhanced Amph-induced behaviors [25,26], and that rats chronically treated with Amph exhibit a long-lasting increase in striatal DA [22], our results establish *C. elegans* as an efficient and inexpensive model to study the long-lasting physiological modifications caused by prenatal exposure to Amph [37].

## 4. Materials and Methods

### 4.1. Worm Husbandry and Strains

Animals were grown at 20 °C in NGM plates seeded with N22 bacteria in noncrowded conditions. The N2 wild-type (Bristol variety), cat-1::GFP (KP1287), and *met-2(n4256)III* and were purchased from the Caenorhabditis Genetic Center (CGC), University of Minnesota, which is funded by NIH Office of Research Infrastructure Program (P40 OD010440).

### 4.2. Embryonic Exposure to Amphetamine

Embryos were released from gravid worms by treating the worms for 5–10 min with a solution containing 10 N NaOH and 1 mL of 10% sodium hypochlorite. After 3 washes with *C. elegans* egg buffer and spinning for 3 min at 1200 rpm, embryos were separated from carcasses in a 30% sucrose gradient and centrifuged at 1200 rpm for 5 min. Floating embryos from the solution meniscus were collected and washed 3 times with sterile water at 1200 rpm. Isolated embryos were then incubated for 15 h with control solution (M9 buffer) or M9 and 0.5 mM Amph (NIH-NIDA, Research Triangle Institute; CSA Schedule 2, drug key 1100-007-003, purity ≥ 95%). Larvae at stage 1 (L1) were washed 3 times in M9 buffer and seeded in NGM plates containing NA22 *E. Coli* bacteria and grown at 20 °C. Approximately 48 h later, when animals achieved the stage of late larvae L4 (young adults), they were tested for SWIP or used for chromatin, mRNA, DA or protein extraction for ChIP, gene expression, ELISA (LDN-13631), or Western blot experiments.

### 4.3. Swimming-Induced Paralysis (SWIP) Assays

In each SWIP trial, 8–16 animals in late larva stage 4 (young adults) were transferred into 40 μL of vehicle (200 mM sucrose, which guarantees physiological osmolarity) with or without 0.5 mM Amph in a single well of a Pyrex spot plate (Thermo Fisher Scientific, Waltham, MA, USA). Paralyzed animals were counted every minute for 10 min using a stereoscope (Carl Zeiss, Inc., Thornwood, NY, USA). The number of paralyzed animals was reported as a percentage of the total number of animals observed in each trial ± standard error.

### 4.4. Chromatin Immunoprecipitation (ChIP) Assays

After 3 washes with M9, late-L4 animals were collected in ice-cold PBS with proteinase and phosphatase inhibitors (Halt^TM^ Thermo Fisher; catalog number 78440). Using a 1 mL pipette, animals were dropped into a 50 mL glass beaker containing liquid nitrogen to flash-freeze worms into ice balls. Frozen worms were disintegrated into powder using a frozen hammer homogenizer. The worm powder was fixed with 1.1% formaldehyde in PBS with proteinase and phosphatase inhibitors (Thermo Fisher) at room temperature for 10 min. The reaction was quenched by adding 125 mM glycine for 5 min. Samples were washed one time with PBS at 4000 *g* for 3 min and two times with Dounce buffer (Sucrose 0.35 M, HEPES-KOH pH 7.5 15 mM, EGTA 0.5 mM, MgCl_2_ 0.5 mM, KCl 10 mM, EDTA 0.1 mM, DTT 1 mM, Triton X-100 0.5%, NP-40 0.25%) at 16,000 *g* for 5 min. Chromatin collected from the top aqueous clear meniscus was fragmented with micrococcal nuclease (10,000 units/mL) to mostly mononucleosomal fragments and incubated with antibodies against H3K4me3 or H3K9me2 (0.025 μg/μL) overnight. The samples were then incubated with 40 μL of prewashed protein A beads plus salmon sperm DNA (4 ng/mL) at 4 °C for 3 h. The immunoprecipitated chromatin was incubated at 65 °C overnight to reverse crosslinking, treated with proteinase K (100 μg/mL) and RNase A (200 μg/mL), and subsequently used as template for qPCR with primers for amplification of specific regions after purification. Three sets of primers were designed for *cat-2*. Primer set 1 amplified the region from position -80 to +24, and primer sets 2 and 3 amplified the regions between positions +583 to +683 and +725 to +840, respectively. Three sets of primers were designed for cat-1. Primer set 1 amplified the region between positions −279 to −132, and primer sets 2 and 3 amplified the regions between positions −130 to +7 and +225 to +335, respectively. Two sets of primers were designed for the housekeeping gene *gpdh*. Primer set 1 amplified the region between positions −27 to +149, and primer set 2 amplified the site in position +1436 to +1615. The reaction procedure was as follows: 95 °C for 3 min, followed by 41 cycles at 95 °C for 10 s and 60 °C for 30 s, and, lastly, 98 °C for 10 s and 65 °C for 31 s, followed by a melt curve from 65 °C to 95 °C. Data were analyzed using the 2^−ΔΔCt^ method. For this method, the average Ct value of act-1, the *C. elegans* actin homolog, was calculated for each sample and used as a control. Then, each Ct value for the genes of interest was subtracted by the actin Ct value average. The product of that subtraction was then subtracted by the average of the “calibrator” (control sample). Finally, the value of the target-calibrator was expressed as 2^−x^, where x represents the target-calibrator value. The final value represents fold change with respect to the “calibrator”.

### 4.5. Quantitative Reverse Transcript PCR (RT-qPCR)

Animals were collected from plates and washed with sterile H_2_O until the suspension appeared clear (no bacteria), followed by two washes in nuclease-free H_2_O. After the final wash, about 3000 worms per condition were used to extract mRNA with the Monarch™ Total RNA Miniprep Kit (NEB, T2010S). NanoDrop 2000c was used to quantify RNA. Briefly, 1 μg mRNA samples were reverse-transcribed into cDNA with the High-Capacity cDNA Reverse Transcription Kit (Applied Biosystems^TM^ Thermo Fisher, catalog number 4368814). Reaction consisted of 3 cycles, priming for 5 min at 25 °C, reverse transcription for 20 min at 46 °C, and RT inactivation for 1 min at 95 °C. Following reverse transcription reaction, the cDNA samples were amplified using PowerUp™ SYBR™ Green Master Mix (ThermoFisher) in a reaction of 20 μL. Reaction procedure was as follows: 95 °C for 3 min, followed by 41 cycles of 95 °C for 10 s and 60 °C for 30 s, and, lastly, 98 °C for 10 s and 65 °C for 31 s followed by a melt curve from 65 °C to 95 °C. Data were analyzed using the 2^−ΔΔCt^ method. For this method, the average Ct value of act-1, *C. elegans* actin homolog, for each sample was calculated and used as control. Then, each Ct value for gene of interest was subtracted by the actin Ct value average. The product of that subtraction was then subtracted by the average of the calibrator (control sample). Finally, the value of the target-calibrator was expressed as 2^−x^ where x represents the target-calibrator value. The final value represents fold change with respect to the calibrator.

### 4.6. Western Blot

Protein lysates were extracted from about 5000 worms with RIPA buffer supplemented with proteinase and phosphatase inhibitors followed by three freeze/thaw cycles in liquid nitrogen. The concentration of proteins was quantified with a BCA protein assay kit (Thermo). Briefly, 60 µg of protein samples per each well were separated in SDS-PAGE and transferred to PVDF membrane (Immobilon^®^-P, Millipore, Burlington, MA, USA) for immunoblotting. Membranes were incubated with primary antibody against human tyrosine hydroxylase (AB152, Millipore) to probe CAT-2 or actin antibody (MAB1501, Millipore), which was used as the loading control. CAT-1 proteins were probed using GFP antibody (A11122, Invitrogen from Thermo Fisher) since samples were collected from animals expressing a CAT-1::GFP fusion protein (KP1287), and, because TH can be fully or partly glycosylated [38], a smeared band was detected. The image was developed with the second antibody conjugated with IRDye (IRDye 800CW, IRDye 680RD, LI-COR). The intensity of protein bands was quantified and analyzed with ImageJ 1.5.3.

### 4.7. Dopamine ELISA Assay

After washing late-L4 worms with H_2_O three times, buffer A (0.01 N HCl, 2 mM EDTA, 4 mM sodium metabisulfite) was added to the worm pellet and immediately frozen to −80 °C. The next day, samples were thawed and homogenized using a tissue grinder. Then, 250 µL of Buffer B (0.01 N NaOH, 2 mM EDTA, 4 mM sodium metabisulfite) was added, mixed very well, and centrifuged at 13,000 rpm for 10 min. After centrifugation, approximately 500 µL of supernatant was collected into a 1.5 mL tube. At this point, DA was quantified using ELISA kits (LDN, Dopamine ELISA^Fast Track^, BA E-6300) as described in the manual. Protein concentration in the supernatant was quantified with a Pierce™ BCA Protein Assay Kit (Thermo Scientific, catalog number 23227).

### 4.8. Statistical Analysis

GraphPad Prism 7 (GraphPad Software, Inc., San Diego, CA, USA) was used for statistical analyses. The statistical significance was determined using 2-way ANOVA with multiple comparison tests and Student’s *t*-tests. The SWIP data passed the Shapiro–Wilk normality test (α = 0.05). Data are reported as averages of at least three independent experiments ± SEM or SD.

## Figures and Tables

**Figure 1 ijms-25-04219-f001:**
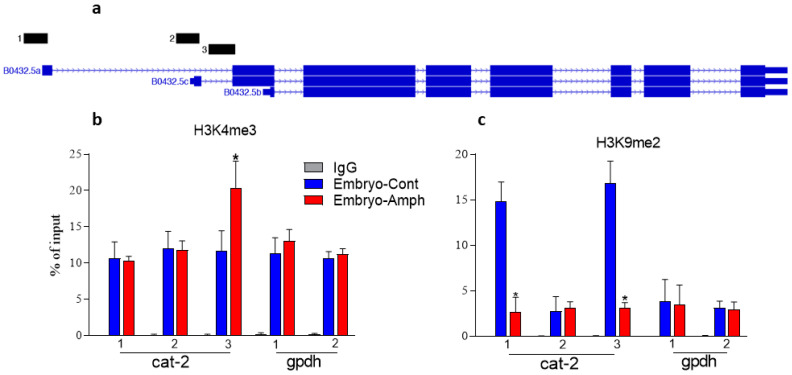
Effects of embryonic Amph exposure at histone marks in the cat-2 and gpdh genes. (**a**) Graphic illustration of the location of primers used to amplify gene regions of *cat-2* for ChIP experiments. (**b**) Antibodies against the H3K4me3 (**b**) and H3K9me2 (**c**) marks were used to immunoprecipitate chromatine samples from adult animals exposed to Amph (red bars) or control (blue bars) during embryogenesis. Both graphs are averages of 3 independent experiments. Statistical analysis was performed using 2-way ANOVA Tukey’s multiple comparison test (* *p* = 0.0001; F(2, 98) = 362 in (**b**) and * *p* = 0.0001; F(2, 59) = 156 in (**c**)).

**Figure 2 ijms-25-04219-f002:**
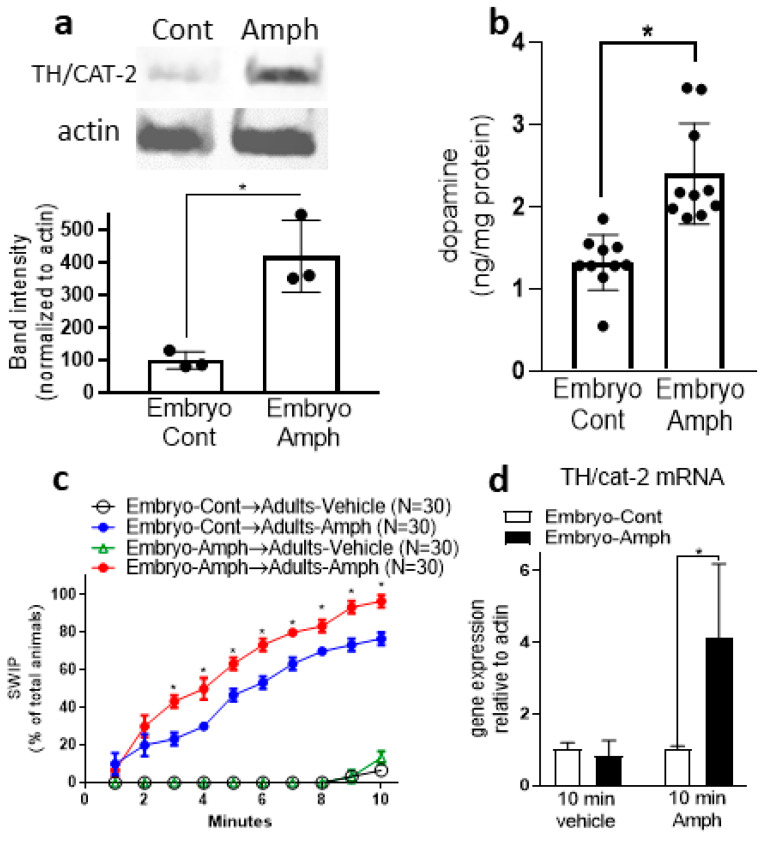
Effects of embryonic Amph exposure on CAT-2 expression and DA levels in adult animals. (**a**) Western blots from whole-worm lysates (* *p* = 0.008; t = 4 and df = 4). A representative blot is shown on top. (**b**) Dopamine ELISA assays from whole-worm lysates (* *p* = 0.0001; t = 4 and df = 18). (**c**) Behavioral assays performed in adult animals treated with 0.5 mM Amph (green triangles and red circles) or control solution (black and blue circles) during embryogenesis and then challenged with vehicle (black circles and green triangles) or 0.5 mM Amph (blue and red circles) (* *p* = 0.0001). (**d**) The *cat-2* mRNA quantification in animals exposed to control or Amph during embryogenesis and then challenged for 10 min with vehicle or 0.5 mM Amph during adulthood (* *p* = 0.01; F(1,8) = 5.8). Each graph represents the average of at least 3 independent experiments. Statistical analysis was performed using Student’s unpaired *t*-test (**a**,**b**) and 2-way ANOVA Tukey’s (**c**) or Sidak’s (**d**) multiple comparisons test.

**Figure 3 ijms-25-04219-f003:**
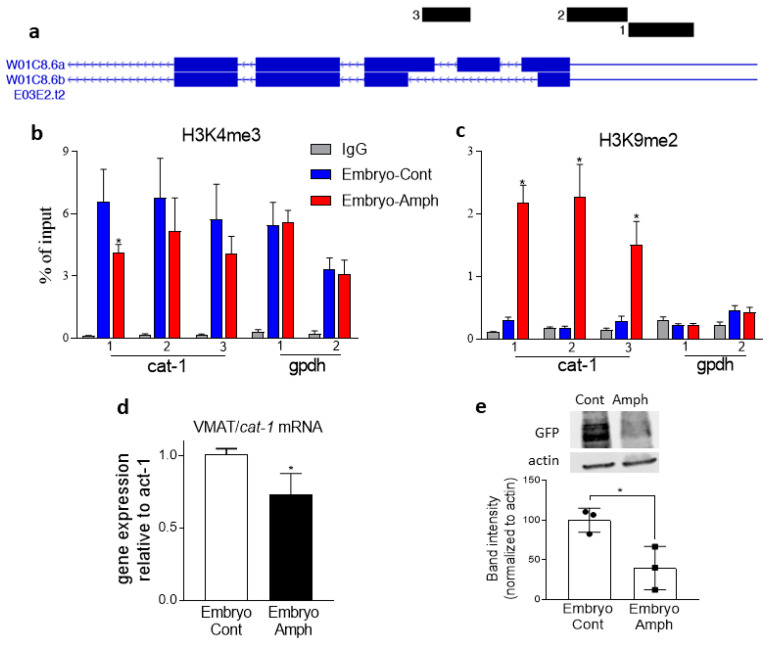
Effects of embryonic exposure to Amph on histone marks and expression of CAT-1. (**a**) Graphic illustration of the location of primers used to amplify gene regions of *cat-1* for ChIP experiments. ChIP assays for H3K4me3 (**b**) and H3K9me2 (**c**) at sites 1, 2, and 3 of the *cat-1* gene or gene sites 1 and 2 of the *gpdh* gene in adult animals exposed to Amph (red bars) or control (blue bars) during embryogenesis. (**d**) *cat-1* mRNA quantification in adult animals that received control (white bar) or 0.5 mM Amph (black bar) during embryogenesis (* *p* = 0.03). (**e**) Western blot assays from CAT-1::GFP expressing animals exposed to control or 0.5 mM Amph during embryogenesis (* *p* = 0.02). Data of each graph are the average of 3 independent experiments. Statistical analysis was performed using 2-wayANOVA Bonferroni’s multiple comparisons test (* *p* = 0.004 in (**b**) and * *p* = 0.0001 in (**c**)) and Student’s unpaired two-tailed *t*-test (**d**,**e**).

**Figure 4 ijms-25-04219-f004:**
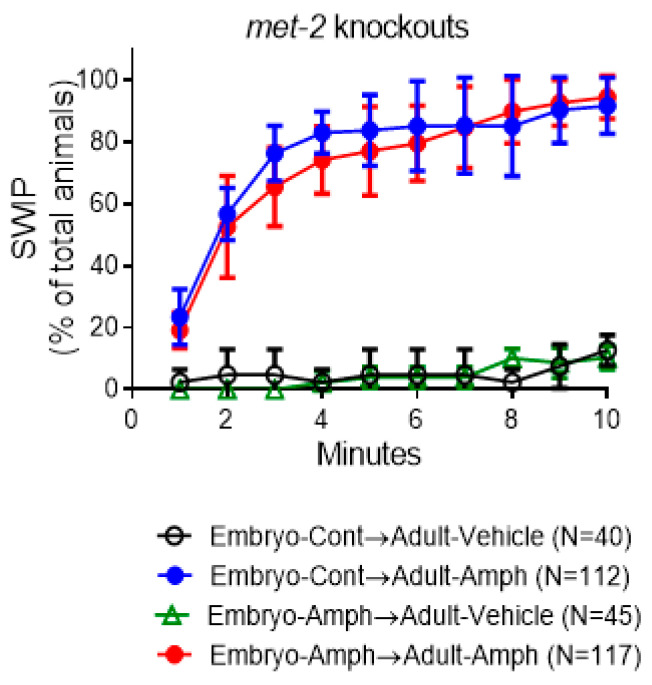
Behavioral assays in *met-2 knockout* animals. *Met-2 knockout* animals were exposed to 0.5 mM Amph (green triangles and red circles) or control (black and blue circles) during embryogenesis and then challenged with 0.5 mM Amph (blue and red circles) or vehicle (black circles and green triangles) during adulthood. Data are an average of 6 independent experiments. Statistical analysis was performed using 2-way ANOVA, Tukey’s multiple comparisons test. No statistical difference was measured between Embryo-Cont→Adult-Amph and Embryo-Amph→Adult-Amph groups (*p* = 0.83; blue and red circles, respectively).

## Data Availability

All data generated in this manuscript are included in this manuscript.

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
