# Peer review of "Amphetamine Exposure during Embryogenesis Alters Expression and Function of Tyrosine Hydroxylase and the Vesicular Monoamine Transporter in Adult C. elegans"

_ijms, 2024, doi:10.3390/ijms25084219_

Round 1

Reviewer 1 Report

Comments and Suggestions for Authors

The authors have used C. elegans to investigate whether exposure to amphetamine during embryogenesis causes changes in the expression and function of two important dopaminergic proteins. They have shown that chronic exposure to amphetamine during embryogenesis increases the expression of CAT-22/TH and decreases the expression of CAT-1/VMAT in adult C. elegans.

The work is well developed and the results appear to be robust. Therefore, I would be happy to recommend publication after the authors address the following points.

My main critical concern is the amphetamine dose. The authors use 0.5 mM, which is about 67 mg/L. How physiologically relevant is this dose for humans? Do the authors have any idea of the amphetamine concentrations in the body of the worms? A quick search in the literature showed that the amphetamine in the blood after lethal intoxication is 0.5-7 mg/L (https://link.springer.com/article/10.1007/BF00200358) (one order of magnitude lower than the dose used in this paper). Blood amphetamine concentrations of 0.27-0.53 mg/L (two orders of magnitude lower than the dose used in this work) have been considered to impair the performance of drivers (https://doi.org/10.1016/j.aap.2005.11.005) What I mean is that I do not think this exposure could be considered chronic, on the contrary, it would be closer to a severe acute or even lethal exposure.

The figure legends are inappropriate. The authors present the results of the figures in the legend and even in the title. For example, "Embryonic Amph exposure increases CAT-2 expression and DA levels in adult animals" should read, in my opinion, something like "Effect of embryonic Amph exposure on CAT-2 expression and DA levels in adult animals". Also, the text should not explain this. For example: "Western blots of whole-worm lysates show a significant increase in CAT-2 probed with the 164 tyrosine hydroxylase antibody in samples from animals exposed to Amph during embryogenesis (*p=0.008; t=4 and df=4)" should be something like: "Western blots of whole worm lysates".

Please include in section four the CAS number of the amphetamine, the supplier, and the purity.

What about the provided files named "ijms-2902257-original-images" and "ijms-2902257-non-published"? Both seem to contain the same information. Also, the text does not cite these files as supplementary material. Please clarify.

Author Response

Previous data from our group showed that 0.3-1mM Amph caused significant SWIP after 5-10 minutes treatments (Safratowich et al 2014). Since 1 mM Amph generated saturating levels of SWIP, in this study, we used 0.5 mM Amph which generates about 60% and 90% SWIP after 5- and 10-minutes, respectively. We don’t know the exact final concentration of Amph in the body of C. elegans during the 0.5mM Amph treatment but, it should be lower than 0.5mM since C. elegans express the monoamine oxidase enzyme, amx-2, which degrades Amph. Moreover, we do not think this concentration is toxic or lethal for C. elegans because:

  1. Our data do not show toxicity after the 15-hours exposure with 0.5 mM Amph during embryogenesis. As matter of fact, all elegans embryos develop in adults with no apparent disfunction and with intact dopaminergic neurons.
  2. Schreiber & McIntire (2010) show that after a 16-hour treatment with methamphetamine, only concentrations equal to or higher than 8 mM were toxic in elegans.

Our use of the word “chronic” refers mainly to the fact that Amph is continuously present during C. elegans’ embryogenesis. This means that neurogenesis and neuronal differentiation occurs under the presence of Amph. To avoid confusion, we have substituted the word chronic with continuous.

We thank Reviewer1 for raising these valid points which we neglected to address in our original submission. Additional text (highlighted in red) has been added in the Discussion section (rows 366-377). 

Figure legends have been changed as recommended by the Reviewer.

Amphetamine information has been added in section 4.

The "ijms-2902257-original-images" and "ijms-2902257-non-published are files generated by the journal. Questions should be directed to the Editor or Journal.

Reviewer 2 Report

Comments and Suggestions for Authors

The authors conducted a study to explore the impact of amphetamine exposure during the embryonic stage in C. elegans, revealing elevated levels of CAT-2/TH and reduced expression of CAT-1/VMAT genes. This study has the potential to uncover the underlying mechanisms of long-lasting physiological and behavioral changes in mammals caused by Amph exposure. However, the data on H3K4me3 and H3K9me2 appears to be confusing, and there seems to be a weak correlation between them.

1.     It is unclear why the cat-2 gene sites 1 and 2 are insignificant.

2.     In Figure 3, Amph exposure during embryogenesis changes histone marks and reduces expression of CAT-1 in H3K4me3 is lower than control and GADPH.

3.     Figure 4 needs full discussion.

Author Response

We thank Reviewer-2 for her/his positive remarks about our manuscript. Additionally, we will clarify the other points raised by the reviewer in the manuscript as follows:

  1. There are three alternative transcription start sites (TSSs) for the cat-2 gene. We designed primer pairs 1, 2 and 3 at the promoters for each of these TSSs (see Fig. 1a for each amplicon’s location). Primer site 1 is upstream of the TSS of the longest transcript (TSS-1). Site 2 is at the TSS of transcript #2, and site 3 is immediately upstream of the TSS of transcript #3. Our ChIP experiments focused on two histone marks at the cat-2 gene, H3K4me3 which is associated with active transcription, and H3K9me2 which is associated with gene repression. Our results show the following:
    1. At primer sites 1 and 2, there is no significant difference in levels of H3K4me3 at sites 1 and 2 in Control vs. Amph treated animals, but site 3 shows an increase in the amount of H3K4me3 in Amph treated animals (Fig. 1b), suggesting that cat-2 transcript #3 might be upregulated.
    2. There is a significant loss of H3K9me2 at primer sites 1 and 3, but not 2 (Fig. 1c). This suggests that cat-2 transcript #1 and #3 might be upregulated because H3K9me2 is associated with gene silencing.
    3. Together, these observations (little gain of H3K4me3 at one site coupled with major loss of H3K9me2) suggest increased transcriptional activity of cat-2 in Amph treated animals relative to control. The increase in cat-2 expression is indeed supported by our Western blot, ELISA and mRNA quantification data shown in Figure 2a, b and d, respectively.
  2. In Figure 3, we are measuring the levels of H3K4me3 and H3K9me2 (Figure 3b-c) at the promoter, TSS, and downstream of cat-1 (see Figure 3a for each amplicon location), while also determining the effects of Amph exposure on cat-1 gene expression and protein levels (Figure 3d-e). We observed a general loss of H3K4me3 (active mark) in sample pre-treated with Amph (red bars). However, this Amph-induced reduction of H3K4me3 was statistically significant only in the amplicon of the promoter region 1. We also observed a significant increase in H3K9me2 (repressive mark) in Amph pretreated animals relative to control in all three amplicons (Figure 3c). This is exactly consistent with the loss of expression of cat-1 as seen by both loss of mRNA and protein expression (Figure 3d-e). On the other hand, these marks are unchanged at the gpdh gene, which is not regulated by Amph and here it is used as a control gene. As expected, gpdh has high levels of gene expression, which is consistent with more H3K4Me3 and low levels of H3K9me2 seen in figure 3. The lack of effect by Amph in the gpdh gene suggests that the long-lasting effects caused by embryonal exposure to Amph do not induce a widespread effect but, it occurs in specific dopaminergic genes (cat-2 and cat-1) only.

We thank Reviewer-2 for her/his insightful remarks, and we agree that further comments needed to be included in our manuscript about data presented in Figure 1, 3 and 4. Our revised manuscript includes further discussion/explanation of the points raised by the Reviewer in the Result and Discussion sections (text highlighted in red).

Round 2

Reviewer 1 Report

Comments and Suggestions for Authors

I am very grateful to the authors for considering and addressing all the points I raised in my first report. In particular, I understand that the issue of acute toxicity raised in the discussion has been addressed. I have no further objections and propose that the manuscript be published.

Author Response

We thank you the Editor for providing very valuable suggestions for our manuscript.

  1. We have clearly stated in the manuscript why 0.5 mM Amph was used in our study (see line 357, highlighted in red) and explain that a lower concentration (0.3 mM Amph) did not cause long-lasting effects.
  2. Rather than adding an extra paragraph, we have discussed in the Discussion the limitations of our study and how the concentration of Amph we used are correlated to those detected in mammals/humans during Amph/Meth binge (line 359-371, highlighted in red).
  3. We have included another reference supporting the data that therapeutical doses of Amph do not affect the health of the offspring (see reference 12 in line 67, highlighted in red).

Reviewer 2 Report

Comments and Suggestions for Authors

The authors addressed all the points that I raised and modified the manuscript. The manuscript is improved and can be accepted. 

Author Response

(The authors gave the same response as above.)
